# Insulin Resistance Is Associated with Reduced Food Odor Sensitivity across a Wide Range of Body Weights

**DOI:** 10.3390/nu12082201

**Published:** 2020-07-24

**Authors:** Maria Poessel, Jessica Freiherr, Kathleen Wiencke, Arno Villringer, Annette Horstmann

**Affiliations:** 1IFB Adiposity Diseases, Leipzig University Medical Center, Philipp-Rosenthal-Str. 27, 04103 Leipzig, Germany, wiencke@cbs.mpg.de (K.W.); horstmann@cbs.mpg.de (A.H.); 2Department of Neurology, Max Planck Institute for Human Cognitive and Brain Sciences, Stephanstr. 1a, 04103 Leipzig, Germany; villringer@cbs.mpg.de; 3Leipzig University Medical Center, CRC 1052A5 ‘Obesity Mechanisms’, Liebigstr. 18, 04103 Leipzig, Germany; 4Department of Psychiatry and Psychotherapy, FAU Erlangen-Nürnberg, University Hospital, Schwabachanlage 6, 91054 Erlangen, Germany; jessica.freiherr@ivv.fraunhofer.de; 5Sensory Analytics, Fraunhofer Institute for Process Engineering and Packaging IVV, Giggenhauser Straße 35, 85354 Freising, Germany; 6International Max Planck Research School on the Neuroscience of Communication, Max Planck Institute for Human Cognitive and Brain Sciences, 04103 Leipzig, Germany; 7Mind Brain Body Institute, Berlin School of Mind and Brain, Humboldt-Universität zu Berlin, 10099 Berlin, Germany; 8Charité—Universitätsmedizin Berlin, 10117 Berlin, Germany; 9Day Clinic for Cognitive Neurology, University Hospital at the University of Leipzig, 04103 Leipzig, Germany; 10International Max Planck Research School on the Life Course, Max Planck Institute for Human Development, 14195 Berlin, Germany; 11Department of Psychology and Logopedics, Faculty of Medicine, University of Helsinki, 00290 Helsinki, Finland

**Keywords:** obesity, odor sensitivity, olfaction, HOMA-IR, insulin resistance

## Abstract

The worldwide obesity epidemic is a major health problem driven by the modern food environment. Recently, it has been shown that smell perception plays a key role in eating behavior and is altered in obesity. However, the underlying mechanisms of this phenomenon are not well understood yet. Since the olfactory system is closely linked to the endocrine system, we hypothesized that hormonal shifts in obesity might explain this relationship. In a within-subject, repeated-measures design, we investigated sensitivity to a food and a non-food odor in the hungry and sated state in 75 young healthy (26 normal weight, 25 overweight, and 24 obese) participants (37 women). To determine metabolic health status and hormonal reactivity in response to food intake, we assessed pre- and postprandial levels of insulin, leptin, glucose, and ghrelin. Odor sensitivity did not directly depend on body weight status/body mass index (BMI) or hunger state. However, we could establish a strong negative mediating effect of insulin resistance on the relationship between BMI/waist-hip ratio and olfactory sensitivity for the food odor. These findings indicate an impact of metabolic health status on sensitivity to food odors. Our results contribute to a better understanding of the mechanisms behind altered smell perception in obesity.

## 1. Introduction

The obesity epidemic is a major health problem that is associated with severe comorbidities such as diabetes, stroke, and cancer [1,2]. Although causal mechanisms and possible treatment approaches are being studied intensively, the occurrence of overweight as defined by a body mass index (BMI) between 25 and 29.9 kg/m^2^ and obesity (BMI > 30 kg/m^2^) has almost tripled within the last 40 years [3]. Currently, obesity has a worldwide prevalence of 13% and overweight of 39% [3]. 

Olfaction in obesity: While the etiology of obesity is multifactorial, one of the main contributing factors driving this rapid increase is the obesogenic environment [4,5]. Our environment is full of energy-rich foods that are advertised via stimuli for all sensory channels. Importantly, the sense of smell plays a crucial role in eating behavior and influences food choice and meal size [6,7]. Individuals with obesity more than those of normal weight are susceptible to external food cues such as food pictures [8,9,10,11] but also food smells [12]. Moreover, individuals with obesity perceive food odors as more pleasant than people of normal weight [13], while being surprisingly less sensitive to odors [14,15]. Previous studies showed that olfactory performance with respect to identification and discrimination of odors, as well as perceptual sensitivity, is low in obesity [15]. The majority of these studies used the Sniffin’ Sticks with a standard olfactory detection threshold (ODT) test, which contains the non-food odors n-butanol, a rather unpleasant odor that naturally forms during fermentation, or phenylethyl alcohol (rose-like odor). Given the body of literature suggesting a negative relationship between BMI and olfactory capacity, researchers postulated diverse mechanisms of metabolic and neural malfunction in obesity [16,17]. However, Stafford and Whittle [13] showed recently, that individuals with obesity compared to those of normal weight show a higher sensitivity to the smell of chocolate. Accordingly, smell capacity might not be quantitatively impaired, but qualitatively altered. Since there is no standardized test for assessing chocolate smell sensitivity, we developed a chocolate odor test kit which is similar to the standard Sniffin’ Sticks in terms of odor concentrations and dilution steps. We decided to use chocolate as a food odor as the smell of chocolate has previously been associated with food cravings [10].

Physiological status and odor sensitivity: In our environment, odors signal the availability of food. Therefore, odor sensitivity may also depend on the hunger status (hungry vs. sated), a notion that has been clearly shown in animals before: rats show enhanced sniffing behavior and higher sensitivity to odors [18] in the fasted when compared to the sated state. In humans, however, results are divergent and show both higher [19] and lower sensitivity [20] in fasted states, or no difference at all [21]. While these studies predominantly used non-food odors, we reasoned that food odors are more relevant in the context of obesity. Thus, we investigated differential effects of food as compared to non-food odors in the hungry and sated state.

Hormones and the olfactory system: Alternatively, metabolic and hormonal differences between study populations might explain controversial results in odor sensitivity. For instance, study populations with obesity class 1 (BMI 30–34.9 kg/m^2^) and class 3 (BMI ≥ 40 kg/m^2^) are afflicted differently by hyperinsulinemia, hyperleptinemia, insulin and leptin resistance or low ghrelin levels [22,23]. Interestingly, Palouzier-Paulignan [24] introduced a complex hormonal and metabolic model that provides a neuroanatomical and -physiological link between the olfactory and endocrine systems. Especially the olfactory mucosa and the olfactory bulbs show a high density of insulin, leptin, and ghrelin receptors [25,26], hormones that are actively involved in signaling and regulating the homeostatic state and modulate odor sensitivity [27]. From this it can be concluded that these hormones might be strong modulators of olfactory perception. Thus, we investigate pre- and postprandial levels of insulin, glucose, leptin, and ghrelin and relate these measures to odor sensitivity.

Obesity and hormones: Individuals with obesity show several hormonal changes that are possibly leading to an altered diet and internal processing of foods. The influence of these hormones on eating behavior and related conditions such as obesity have been intensively studied in recent years [28]. While orexigenic hormones such as ghrelin and adiponectin stimulate appetite and food intake, anorexic hormones such as insulin and leptin induce satiety and regulate long-term energy homeostasis [29]. Obesity is associated with higher levels of insulin and leptin [30,31], while sensitivity to these hormones is reduced [32,33]. In addition, the plasma total ghrelin levels and ghrelin reactivity are lower in individuals with obesity when compared to those of normal weight [22,23,34]. Typically, ghrelin decreases after eating in healthy normal weighted individuals [35]. Since the ghrelin response to food intake is blunted in obesity, ghrelin might act independent of attenuated physiological needs in obesity, because ghrelin level does not properly decrease after a meal. Accordingly, individuals with obesity might experience unaffected high appetite after eating. For ghrelin, an acylated (active) and unacylated (inhibiting) form exists [36]. Of particular interest is the elevated ratio of acylated (AG) to unacylated ghrelin (UAG) in obesity [37]. It is pivotal for maintaining weight balance [36], since an elevated AG/UAG ratio could reflect the lower level of UAG and thus be responsible for a consistently high appetite and urge to eat even after a meal. 

Summary: Within this study we aim to explore whether the effects of weight status and hunger state on olfactory sensitivity are mediated by endocrine changes in obesity. Based on current evidence, we assumed that participants of normal weight would outperform those with obesity for the non-food odor and vice versa for the food odor. Second, we hypothesized that while individuals of normal weight have a lower odor sensitivity to food odors when they are sated, individuals with obesity would not show this change. Third, we assumed that the endocrine profile of participants with obesity is characterized by high insulin resistance, high leptin levels, elevated AG/UAG ratio, and low total ghrelin levels. We further expected that the endocrine profile would mediate the relationship between BMI and olfactory sensitivity. 

## 2. Materials and Methods 

### 2.1. Subjects

The sample consisted of 84 participants. We excluded *n* = 9 due to stuffed noses, poor veins, and insufficient intake of calories/satiation in the sated condition. Thus, data from 75 participants were analyzed. All participants were recruited from the Max Planck Institute database. They were aged between 18 and 35 years (27.2 ± 3.7 years) to exclude age effects on olfactory performance [38] and their BMI ranged from 18.8 to 44.2 kg/m^2^. Exclusion criteria included smoking, recent history of smoking (<3 years abstinence), vegetarian/vegan diet, allergies, alcoholism (reported intake > 5 times per week), pregnancy/breastfeeding, nose surgery except childhood polypectomy, and history of neurological/psychiatric disorders. Participants provided written informed consent. The study was carried out in accordance with the Declaration of Helsinki. The study protocol was approved by the Ethics Committee of the University of Leipzig (170-16/ek-25042016).

### 2.2. Study Design

On two consecutive days, the study participants came to their testing at the same time each day. They fasted overnight (approx. 12 h) and received a meal on the first or second day in pseudo-randomized order (Figure 1). The meal consisted of a cereal-fruit-smoothie (either banana or wild berry) with 25% of the participants’ daily energy requirement determined by an interview about their physical activity level and body weight/height. All participants were screened for olfactory function via the short form [39] of olfactory identification test. They underwent a medical examination to assess body weight, height, waist, and hip circumference as well as several interviews. On both test days, participants rated intensity, pleasantness, and familiarity of the odorants that were applied for odor sensitivity testing on visual analogue scales [40] (How strong or intense is this odor?/0 = very weak, 10 = very strong; How pleasant/unpleasant is this odor?/−5 = unpleasant, +5 = pleasant; How familiar is this odor?/0 = unfamiliar, 10 = familiar). We collected blood samples on both test days in the morning and 60 min after meal onset/break without meal. Thirty minutes after meal onset/break without meal, participants performed ODT tests for the three odors (food pleasant: chocolate; non-food pleasant: grass; and non-food standard: n-butanol) in pseudo-randomized order. 

### 2.3. Questionnaires and Interviews

Depressive symptoms were assessed using the Beck Depression Inventory (BDI) [41] in a paper pencil form before olfactory testing to directly control for acute suicidal tendencies and exclude participants with more than mild depressive symptoms (>score 18). Additionally, before starting the study protocol, the participants were face to face interviewed about their smoke status as an additional control to the telephone interview to exclude people according to our exclusion criteria on smoking. The conducted smoking interview was previously used in the Leipzig Life Study, and included questions on smoking behavior in the past and present, on the beginning and duration of smoking, on breaks and passive smoking [42]. Women were further interviewed to obtain information about their menstrual cycle, because sensitivity to odors is increased in follicular phase of the cycle/under oral contraceptive and decreased in luteal phase [43,44].

### 2.4. Testing of Olfactory Performance

All odorants were presented in commercially available felt-tip pens (Sniffin’ Sticks; Burghart Instruments, Wedel, Germany). For the testing of olfactory sensitivity, the ODT test-kit from the Sniffin’ Sticks test battery was implemented for the standard non-food odor. The kit consists of a geometric series of sixteen dilutions of n-butanol. We further developed a test-kit for a sweet, high fat food odor (chocolate). As a pilot study indicated that the standard test odor n-butanol is perceived as rather unpleasant, we decided to develop a second test-kit with a pleasant non-food odor (freshly mown grass). We therefore filled the odorless tampons of blank Sniffin’ Sticks with either chocolate odor (chocolat noir, code:1130/4, Givaudan) or grass (cis-3-hexen-1-ol, code: H12900-1OG, Aldrich). Based on the common n-butanol test we developed a similar geometric series of sixteen dilutions for each odor. To obtain similar perceived intensity of the three odors, chocolate pens were filled with highest 1% and lowest 31 ppm and grass pens were filled with a highest odor concentration of 4% and the lowest of 1.22 ppm, both in a similar ratio of 1:2 (pilot study, *n* = 10). We then applied the short single staircase procedure as described in Poessel et al. [45] to determine odor thresholds.

### 2.5. Blood Collection

Blood samples were collected after 12 h of overnight fasting in vacutainer tubes treated with ethylenediaminetetraacetic acid (EDTA) for serum (insulin, leptin) or with aprotinin (AG, UAG, and total ghrelin). Serum was collected in 7 mL Sarstedt monovettes. After 30 min, tubes were centrifuged at 4000 r.p.m. at 4 °C. The serum was separated in two 1 mL tubes (Eppendorf Safe-Lock Tubes) and immediately frozen at −80 °C. Blood for glucose determination was collected in 2.7 mL glucose monovettes and treated as serum tubes. Acylated and unacylated ghrelin was measured by ELISA easy sampling kit (Bertin Pharma, Montigny-le-Brettoneux, France) and total ghrelin was measured by ELISA kit (Millipore Corporation, MA, USA). Blood for ghrelin was collected into 5 mL aprotinin tubes to avoid hormone degradation, which was then put on ice immediately at 2–4 °C for max 15 min. Tubes were centrifuged for 10 min at 3500 rpm at 4 °C and after pipetting in two 0.5 tubes immediately stored at −80 °C.

### 2.6. Data Analysis

R version 3.4.3. within RStudio [46] and SPSS (version 22.0, SPSS Inc., Chicago, IL) were used for statistical evaluation. We used BMI as a grouping variable (normal weight: BMI 18.4–24.99 kg/m^2^, overweight: BMI 25.0–29.99 kg/m^2^, and obese: BMI > 30 kg/m^2^) or as a continuous variable. We used waist–hip ratio (WHR) as an additional measure of weight status that is more related to metabolic health and reflects visceral fat status. As an estimate of insulin resistance, we used the homeostasis model assessment-insulin resistance (HOMA-IR, [47]), applying the formula: HOMA-IR = glucose (mmol/L) × insulin (pmol/L)/135. The α-level was set at 0.05. We used Bonferroni correction to adjust the α-level for multiple testing. Whenever statistical assumptions for parametric testing were violated, we applied non-parametric robust tests. We defined outliers as values below or above 2.2 interquartile range from the samples lower or upper quartile [48]. We used standardized values (z-transformation) whenever we directly compared the three odorants, because they slightly differed in perceived intensity. To compare food vs. non-food odor condition, we used chocolate as food and only grass as non-food odor as they were perceived as similarly pleasant. To depict change over time we applied a residualized change model instead of the difference score model to avoid the regression to the mean phenomenon [49]. We used multivariate ANCOVA to depict differences between BMI groups (=between subject variable) for the three ODT values as within subject variables, using “sex” as a covariate. Assumption tests showed homogeneity of variances/covariances, as assessed by Box’s M test. There were no univariate/multivariate outliers, as assessed by standardized residuals greater than ± 3 standard deviations/Mahalanobis distance. Residuals were normally distributed for chocolate and n-butanol, but not for grass odor sensitivity. However, we decided to apply the one-way MANCOVA since it is robust to deviations from normality. Further, we applied a two-way mixed repeated measures ANCOVA to depict the influence of hunger state on olfactory performance. Having a 3 × 2 × 2 design, weight groups (normal weight, overweight, and obesity) were used as between-subject factor; ODT scores (food vs. non-food) and hunger status (hungry vs. full) as within-subject factors and “sex” as a covariate. There were three outliers. Thus, we performed our analysis with and without the outliers. Since results of both analyses did not differ, we decided to keep the outliers within our analysis. ODTs were normally distributed in the hungry, but not in the sated state. Since ANOVAs are robust to deviations from normality, we decided to run this analysis anyway. We used mediation analysis to explore the processes that might underlie the relationship between BMI and odor sensitivity by introducing a third variable: the hormonal status. Unstandardized indirect effects were computed for each of 10,000 bootstrapped samples, and the 95% confidence interval was computed by determining the indirect effects at the 2.5th and 97.5th percentiles. Additionally, we repeated the mediation using WHR as independent variable to investigate the influence of visceral fat on olfaction. For a post-hoc test we grouped our data according to insulin resistance (NH: normal HOMA-IR ≤ 1; EH: elevated HOMA-IR > 1). Since there is no absolute value for HOMA indices and no fixed classification of “normal” and “abnormal” values, our grouping relies on recent data from Lee et al. [50].

## 3. Results

Participant characteristics, hormonal parameters that are associated with metabolic health, ODT scores, and odorant ratings are depicted in Table 1. BDI scores indicated no depressive symptoms in all participants. All participants were non-smokers. The weight groups differed in their endocrine profile. Participants with overweight and obesity showed significantly higher HOMA-IR scores and leptin levels than participants with normal weight. Total ghrelin was lower in participants with obesity when compared to participants with normal weight/overweight. AG/UAG ratio did not differ between groups (for more detail see Appendix A). Odorant ratings did not differ between groups.

### 3.1. Main Hypotheses

#### 3.1.1. Hypothesis 1: General Olfactory Sensitivity Differs Between Weight Groups Depending on Odor Quality

We determined the effect of body weight status on olfactory sensitivity for chocolate, grass, and n-butanol with a one-way MANCOVA. We could show that there was no statistically significant difference between the weight groups on odor thresholds, F(2,138) = 0.004, *p* = 0.881, Wilks’ lambda = 0.967, and partial eta^2^ = 0.017 (Figure 2). 

#### 3.1.2. Hypothesis 2: Olfactory Sensitivity Depends on Hunger State 

A two-way mixed repeated-measures ANCOVA was run to determine the effect of hunger state on odor sensitivity for the food and non-food odor depending on weight status. There was no main effect of hunger state (F(1,64) = 0.189, *p* = 0.665, and rm-ANCOVA), meaning that being hungry or sated had no influence on odor thresholds across groups (see Appendix A for sanity checks on the applied hunger modulation). Additionally, there was no between subjects effect for weight status (F(2,64) = 0.559, *p* = 0.575, and rm-ANCOVA), meaning that being hungry or sated did not differently affect olfactory performance in the three weight groups (Figure 3). Finally, there was no significant interaction between odor, hunger state, and BMI group on ODT score (F(2,66) = 1.913, *p* = 0.156, and partial eta^2^ = 0.055).

#### 3.1.3. Hypothesis 3: Hormones Mediate Olfactory Performance 

##### General Odor Sensitivity

The relationship between BMI and odor sensitivity for chocolate was mediated by metabolic health parameters. As Figure 4A,C illustrate, the standardized regression coefficients between BMI/WHR and metabolic health indicators were significant for IR (insulin resistance) as assessed by HOMA-IR score and leptin levels (a-path), as were the standardized regression coefficients between IR and odor sensitivity (b-path). The standardized indirect effect between BMI and odor sensitivity via IR was −0.256 (CI −0.689 and −0.026) and for WHR −0.241 (CI −0.689 and −0.026). There was no direct or indirect relationship between BMI and odor sensitivity for the non-food odor condition (Appendix A).

##### Change in Odor Sensitivity in Response to a Meal

The relationship between BMI and the change in odor sensitivity after meal intake is indirectly predicted by hormonal parameters that are related to metabolic health. As Figure 4A,D illustrate, the standardized regression coefficients between BMI/WHR and hormones were significant for IR and leptin. The b-path was significant for IR, leptin, and total ghrelin in the BMI model and for IR only in the WHR model. The standardized indirect effect was significant for IR –0.443 (CI –0.546 and –0.214) and leptin −0.441 (CI −0.512 and −0.112).

### 3.2. Post Hoc Analyses 

To understand the role of insulin resistance in the interplay of olfaction and obesity, we performed a post hoc analysis of the hypothesis 2 model using IR as an additional covariate. Interestingly, we could now show a main effect of BMI group for the food odor *F*(2, 65) = 3.303, *p* = 0.043, and partial η^2^ = 0.093 (Figure 5). Post hoc comparisons showed that individuals with obesity outperformed those of normal weight when IR was controlled (mean difference = −1.293, SE = 0.391, *t* = −3.309, and *p* = 0.005; Tukey test). 

Finally, for another post-hoc analysis, participants were classified into two groups according to their insulin resistance: optimal HOMA-IR ≤ 1 (*n* = 41) and elevated HOMA-IR > 1 *(n* = 33). We performed a multivariate ANOVA using IR-group as group variable and olfactory sensitivity as dependent variable. Olfactory sensitivity was higher in the optimal HOMA-IR group (food: *M* = 7.9, *SD* = 1.3; non-food: *M* = 10.4, *SD* = 1.7) when compared to elevated HOMA-IR group (*M* = 7.1, *SD* = 1.5; non-food: *M* = 9.7, *SD* = 2.2). The difference was significant for the food odor condition, ODT food: *F*(1,74) = 8.608, *p* = 0.005, and partial η^2^ = 0.108; ODT non-food: *F*(1,74) = 2.613, *p* = 0.110, and partial η^2^ = 0.035. To sum up, BMI seems to be positively associated with olfactory sensitivity when controlling for IR. IR is negatively associated with olfactory sensitivity in the food odor condition independent of BMI (see Figure 6).

## 4. Discussion

The aim of the study was to shed light on the complex relationship between obesity and smell perception. Hence, we examined different modalities that could influence smell perception in participants with normal weight, overweight and obesity. We directly compared sensitivity to food and non-food odors and included the influence of being hungry or sated on smell sensitivity. Additionally, we assessed endocrine parameters to evaluate the metabolic health status of our participants. Since it has been shown that olfaction is directly influenced by hormones that play an important role in obesity, we related metabolic health status to our measurements of olfactory sensitivity. 

In accordance with recent literature on the relationship between the olfactory and endocrine systems [24,51,52], our study shows for the first time that there is a negative indirect effect of BMI on odor sensitivity for chocolate through the metabolic health parameter insulin resistance (IR). This means that the higher the BMI the lower odor sensitivity via IR (Figure 3). With the applied mediation model, we could, as expected, show positive effects of BMI on IR and circulating leptin levels, i.e., the higher the BMI the higher IR and leptin. Furthermore, we could show that there is a negative effect of IR on odor sensitivity while controlling for BMI, i.e., the higher IR the lower odor sensitivity independent of BMI. Moreover, we could show for the first time that WHR as a measure of visceral fat mass has a direct effect on food odor sensitivity and that this effect is again mediated by IR. These findings promote suggested metabolic and hormonal mechanisms of altered smell perception in obesity [15,16]. In fact, our results offer a possible explanation for controversial findings in smell sensitivity, since most studies did not include hormonal parameters as control-variables and most likely included participants of different metabolic health status. Interestingly, the one study providing evidence of a higher smell sensitivity in obesity is the only one that examined very young and class 1 obese participants [13]. Other studies examining this relationship also included older and class 1–3 obese participants, e.g., [51,53]. (Appendix A for comparison). In the current study, the obese sample consists of 17 class 1 obese participants (BMI 30–35 kg/m^2^), five class 2 participants (BMI 35–40 kg/m^2^), and two class 3 participants (BMI > 40 kg/m^2^). In this respect, we think it is not surprising that olfactory ability is not yet generally decreased in our sample with obesity.

In addition to this general effect, we have also established a strong mediating effect of metabolic health markers on the relationship between BMI and the change in odor sensitivity in response to meal intake. We expected that odor sensitivity decreases in response to food intake reflecting less need to search for nutrients in the environments as has been reliably shown in animal models [18,54,55]. We assumed that this mechanism may be changed by poor metabolic health, and thus insufficient modulation of central odor sensitivity in obesity. Namely, we expected that while participants with normal weight would show a significant decrease in response to meal intake, participants with obesity might show similar odor sensitivity in the hungry and sated state in accordance with most recent literature on low reactivity of sensory (odor intensity) and metabolic (ghrelin reactivity) systems in response to changing hunger states in obesity [52]. In contrast with our expectations, we detected a negative indirect effect of BMI on change in odor sensitivity through the mediator IR, meaning that the higher the BMI together with high IR, the more negative is the change in odor sensitivity. A negative change score represents a decrease in odor sensitivity, while a positive score represents an increase in odor sensitivity in the sated when compared to the fasted state. Hence, odor sensitivity for chocolate decreases in metabolically impaired individuals with a high BMI after a meal, while no change in odor sensitivity is observed in metabolically healthy participants. However, we show a positive direct effect of BMI on change in odor sensitivity for chocolate depending on hunger state independent of metabolic health parameters. Thus, the effect of BMI on the change in odor sensitivity in response to a meal is positive after removing the effect of metabolic health (= adjusting for the mediators). This means that a high BMI is associated with an increase in odor sensitivity in the sated state, but only in metabolically healthy individuals. Further applied post hoc analyses underpinned this finding. We could show that participants with obesity outperformed those of normal weight while controlling for IR, but this effect was not present in the same design without IR as a covariate (original Hypothesis 2). Thus, individuals with obesity, who are metabolically healthy might have higher sensitivity for food odors in the sated state, while people affected by high HOMA-IR values independent of body weight status show lower olfactory sensitivity for food odors. Since we find these effects in the food odor condition only, our results support the proposed idea from Riera and colleagues [56], that reduced olfactory input protects against intake of high caloric diet by mimicking reduced odor sensitivity after meal intake. This might also reflect a specific reaction of our sensing system to food odor cues in response to our hunger state. In contrast to previous findings [51,52], we could not confirm any association between ghrelin and odor sensitivity in our data set. However, we found a weakly negative correlation between total ghrelin and BMI as has been previously reported [34].

### Limitations

Although many important potential confounders such as menstrual cycle phase in women (see Appendix A) are well controlled in the current study, there are limitations to consider as well. First, the hormonal profile in the current study is limited and may expand in future studies, assessing new hormonal and inflammatory markers that are possible targets to explain the relationship between obesity/metabolic health and olfactory perception. Secondly, the reliability in olfactory testing is generally rather low, possibly because of the susceptibility of our olfactory sense to many external factors such as smoking [57] and menstrual cycle [43,44]. While we attempted to counteract those influences by controlling for several confounding factors (smoking, season, cycle phase, and hunger), we were unable to address all possible confounders satisfactorily. Especially, since our measures rely on repeated sensitivity measurements, conclusions must be drawn with caution, because test–retest reliability ranges only from 0.43–0.86 in ODT testing [58,59,60,61]. However, we tried to counteract this weakness by randomizing the test day order. In addition, an unchanged sensitivity for chocolate could be due to the dissimilarities between the ODT test odor (chocolate) and the standard meal (cereal smoothie), as the greatest change would be expected when sensory specific satiety occurs [62]. Moreover, we did not include a direct measure of eating behavior. Hence, we cannot make a statement about the effect of odor sensitivity on food intake in everyday life.

## 5. Conclusions

This study supports recent literature on a close relationship between the olfactory system and metabolic parameters in obesity [51,52]. To our knowledge, it is the first study that shows a strong mediating effect of the metabolic health parameter insulin resistance on the relationship between BMI and odor sensitivity for chocolate. Thus, it provides further understanding of the pathophysiological mechanisms that might underlie altered smell perception in obesity.

## Figures and Tables

**Figure 1 nutrients-12-02201-f001:**
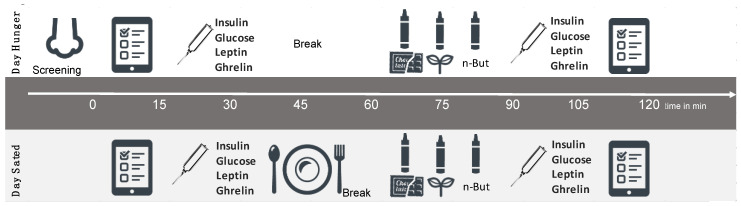
Study Design.

**Figure 2 nutrients-12-02201-f002:**
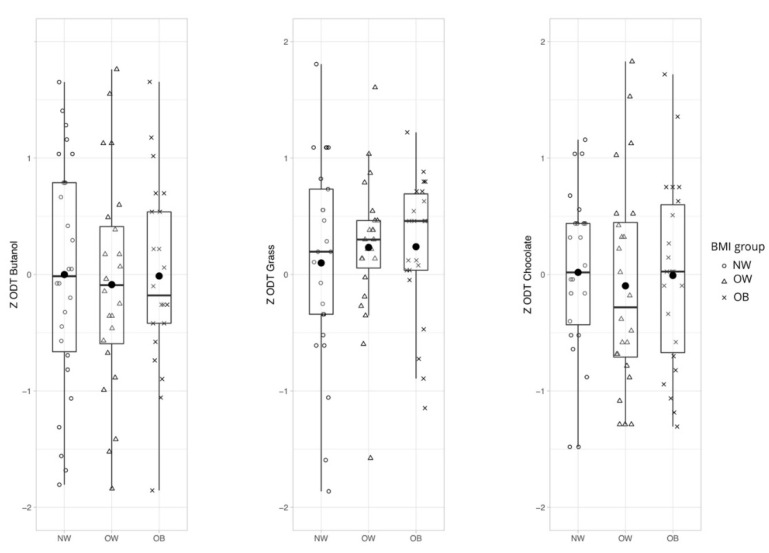
Hypothesis 1: General odor sensitivity for chocolate, grass, and n-Butanol in different weight groups. No statistically significant difference between normal weight, overweight, and obese participants on odor thresholds after controlling for sex, F(2,138) = 0.004, *p* = 0.881. Abbreviations: NW—normal weight; OW—overweight; OB—obese; and ZODT—z-standardized score for odor thresholds.

**Figure 3 nutrients-12-02201-f003:**
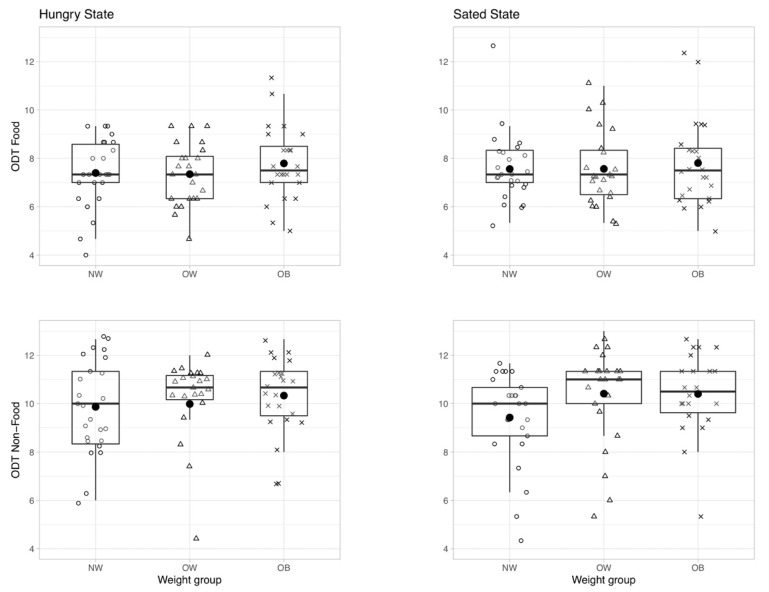
Olfactory detection threshold (ODT) for food (top) and non-food odor condition (bottom) according to hunger state: hungry (left) vs. sated (right); depicted for different weight groups (NW—normal weight, OW—overweight, and OB—obese). ODT ranges from score 0—highest odor concentration (= low odor sensitivity) to 16—lowest odor concentration (=high odor sensitivity). No significant main or interaction effect.

**Figure 4 nutrients-12-02201-f004:**
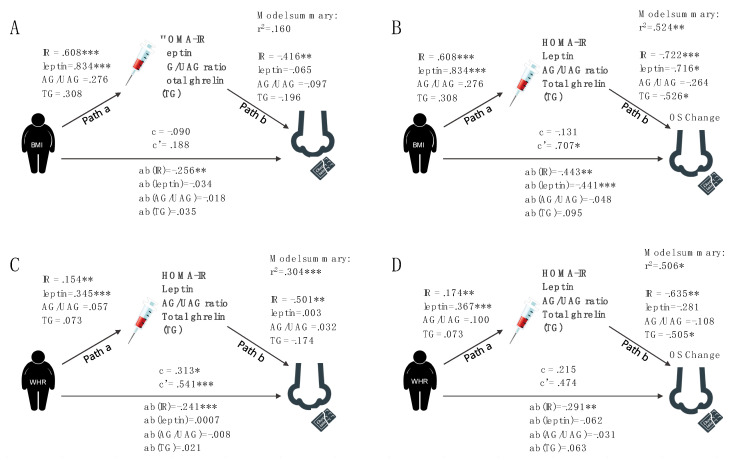
Mediation analysis for odor sensitivity to chocolate. (**A**) Mediation analysis for general chocolate odor sensitivity. Path a—represents the relationship between BMI and hormonal parameters; b—represents the relationship between odor sensitivity and hormonal parameters; c—represents the total effect (direct + indirect effect); c’—represents the direct relationship between BMI and odor sensitivity; ab—represents the indirect effect of hormonal parameters on the relationship between BMI and odor sensitivity. (**B**) Mediation analysis for the change in chocolate odor sensitivity (OS change) in response to meal intake. (**C**) Mediation analysis for chocolate odor sensitivity using waits–hip ratio (WHR) instead of BMI as independent variable. (**D**) Mediation analysis for the change in chocolate odor sensitivity (OS change) in response to meal intake using WHR as independent variable.

**Figure 5 nutrients-12-02201-f005:**
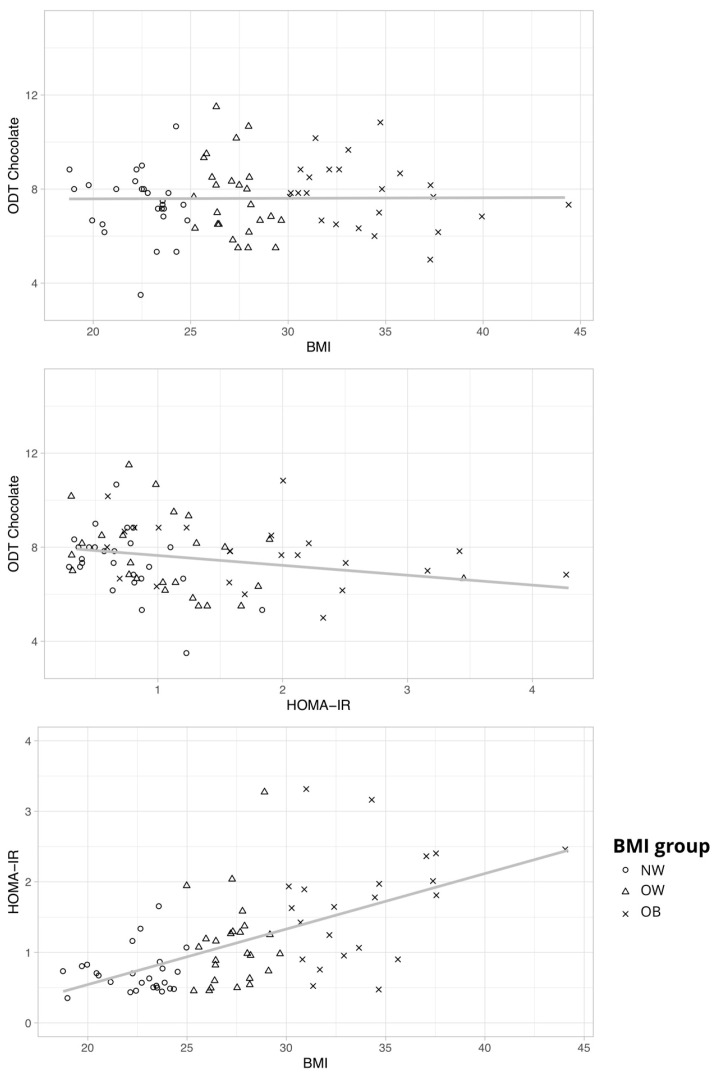
Scatterplots with regression lines to depict the relationship between chocolate odor sensitivity (ODT Chocolate), BMI, and HOMA-IR.

**Figure 6 nutrients-12-02201-f006:**
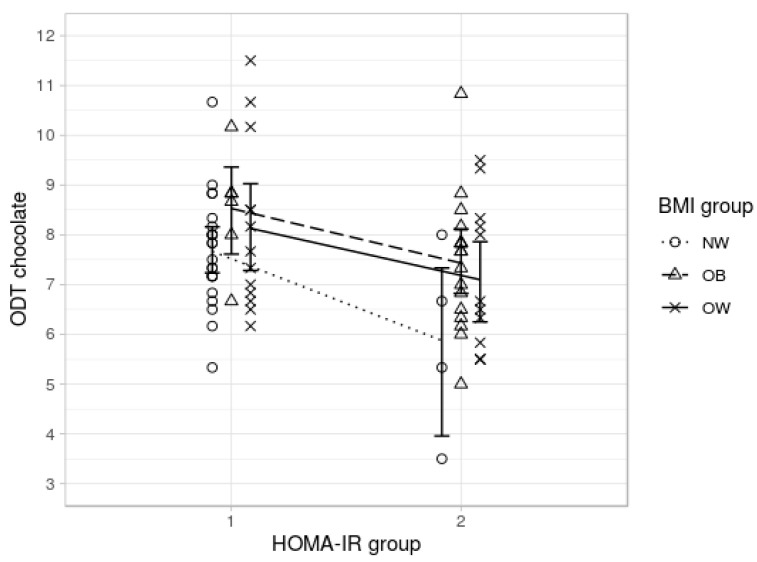
Chocolate odor sensitivity (ODT chocolate) for optimal (HOMA-IR < 1) and elevated HOMA-IR group (HOMA-IR > 1) with separate lines for BMI groups (NW—normal weight, OW—overweight, and OB—obese).

**Table 1 nutrients-12-02201-t001:** Participant characteristics, metabolic and endocrine profiles, odor detection thresholds, and odorant ratings.

	Total	Normal Weight (NW)	Overweight (OW)	Obese (OB)	*p*-Value
	(75, 37 females)	(*n* = 26, 14 females)	(*n* = 25, 12 females)	(*n* = 24, 11 females)	
***Characteristics***					
Age (years)	27.2 ± 3.7	26.1 ± 2.7	27.3 ± 1.3	27.7 ± 4.4	*p* = 0.142 ^a^
BMI (kg/m^2^)	27.8 ± 5.3	22.4 ± 1.7	27.3 ± 1.3	34.1 ± 3.5	*p* < 0.001 ^a^
BDI sum	4.0 ± 3.6	2.9 ± 3.0	4.5 ± 4.1	4.5 ± 3.4	*p* = 0.169 ^a^
Passive Smoke	3.0 ± 8.5	4.8 ± 13.9	2.1 ± 2.7	2.1 ± 3.5	*p* = 0.443 ^a^
***Metabolic profile***					
HOMA-IR	1.18 ± 0.79	0.71 ± 0.30	1.11 ±0.63	1.78 ± 0.94	*p* = 0.019 ^b^, *p* < 0.001 ^c^, *p* = 0.018 ^d^
Leptin	15.59 ± 17.88	6.19 ± 3.92	12.65 ± 13.1	26.77 ± 23.25	*p* < 0.001 ^a^
Total ghrelin	571.53 ± 218.17	604.35 ± 238.32	626.46 ± 186.43	478.75 ± 204.12	*p* = 0.926 ^b^, *p* = 0.097 ^c^, *p* = 0.044 ^d^
AG/UAG ratio	19.92 ±14.21	21.39 ± 16.25	16.22 ± 8.67	22.19 ± 16.17	*p* = 0.278 ^a^
***Odors***					
**N-butanol**					
Pleasantness		3.72 ± 1.80	3.23 ± 1.71	4.14 ± 2.03	ns ^a^
Intensity		7.28 ± 1.78	7.52 ± 1.80	6.67 ± 1.88	ns ^a^
Familiarity		5.72 ± 2.51	6.00 ±2.79	5.16 ± 2.65	ns ^a^
**Chocolate**					
Pleasantness		8.03 ± 1.46	7.76 ± 2.04	7.55 ± 1.54	ns ^a^
Intensity		7.99 ± 1.70	7.79 ± 1.64	7.79 ± 1.45	ns ^a^
Familiarity		8.61 ± 1.32	8.80 ± 1.24	8.26 ± 1.65	ns ^a^
**Grass**					
Pleasantness		7.31 ± 1.55	7.46 ± 1.92	6.58 ±2.51	ns ^a^
Intensity		7.73 ± 1.63	7.80 ± 2.05	7.64 ± 1.70	ns ^a^
Familiarity		7.69 ± 2.10	7.90 ± 2.83	7.35 ± 2.71	ns ^a^

Abbreviations: BMI—body mass index; BDI—Becks Depression Inventory; Passive Smoke—passive smoking hours per week; HOMA-IR—homeostatic model assessment—Insulin Resistance; AG/UAG ratio—ratio of acylated to unacylated ghrelin ^a^—between all groups; ^b^—between NW and OW group; ^c^—between NW and OB group; ^d^—between OW and OB group.

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
