# Peer review of "Insulin Resistance Is Associated with Reduced Food Odor Sensitivity across a Wide Range of Body Weights"

_nutrients, 2020, doi:10.3390/nu12082201_

Round 1

Reviewer 1 Report

This is a novel report providing some evidence of the link between metabolic determinants and olfactory sensitivity. The authors clearly stated the present gap in knowledge and how the aims of this study attempted to address that. The presentation of data is of high quality and the discussion of the findings is presented in methodical and well organized manner. The discussion of limitations is also commendable, as there are many other factors not addressed in this report that could affect odor sensitivity (e.i. environmental factors as well as genetic determinants).

Author Response

We thank the reviewer for his positive evaluation of our manuscript. We also think that the sense of smell is very susceptible to interference and are pleased that the reviewer appreciates our discussion on this point.

Reviewer 2 Report

The manuscript presents a significant research study to investigate odor sensitivity for a chocolate odor and a grass odor in the hungry and sated state in young normal weight, overweight, and obese groups. Previous studies have controversial findings in odor sensitivity in obesity groups compared to normal weight groups; some found obesity groups showed low olfactory performance and others found obesity groups showed a higher sensitivity compared to normal weight group. This paper has a possible explanation for these controversial findings; there is a strong mediating effect of IR on the relationship between BMI and odor sensitivity for a food odor, chocolate. I only have a few minor comments.

Abbreviations/acronyms: there were several abbreviations/acronyms the first time they were used in the abstract and the text without defining them. Or some of the abbreviations/acronyms were introduced later after used several times in the text. A few examples here, HOMA-IR or BMI/WHR in the abstract or PEA (Line 60) or EDTA (Line 160) in the text.

Line 49: please define obesity and overweight based on the body mass index.

Line 59-60: it would be great to include odor descriptions for n-butanol and PEA. Readers may be curious what kinds of non-food odors were used in the previous studies, knowing that the current study used ‘grass’ as a non-food odor.  

Line 64-66: please re-write this sentence (Since…) because it was not easy to understand right away and it seemed weak justification why chocolate was used as a food odor. So the authors developed two test kits for a food odor and a non-food odor to directly compare those two odors with the same methods, right? Why chocolate was chosen as a food odor? The test kit could be a model to develop to test sensitivity for other odors?   

Line 132-134 (as well as several…): same information as the Questionnaires and Interviews section (Line 170-180). Maybe the section of Questionnaires and Interviews could be moved up here. It may be easier to follow because these questionnaires and interviews were performed before olfactory performance testing. Or Under the Study Design, questionnaire and interviews, testing of olfactory performance and blood collection could be subheadings. Just suggestions..

Line 135-136: ‘.. of the pens containing the highest concentration of chocolate, grass, and n-butanol on visual analogue scales.’ This sentence was placed before explaining the pens the authors developed. Thus, as a reader, I was confused what these pens are and what the highest concentrations mean. Further, please explain more about the visual analogue scale. What was the length of the scale (maybe 10 cm..)? How it was labeled? Also, please indicate this in the Table 1 (Line 223-) as a note to explain how the pleasantness, intensity and familiarity were measured on a what scale.  

Line 141: Figure 1, were there questionnaires or interviews after the blood collection? I can’t find in the text, but I see the table pc icon after the blood collection.

Line 181-214: It would be helpful if the covariates and fixed effects were specified when ANOCOVA or MANCOVA were explained.

Line 223: Table 1 – it seems that NW group had a relatively low BDI and high Passive Smoke compared to OW and OB groups even though there were no significant p-values found. Any chance these may affect your results?

Line 273: Figure 4 – the font size in the figure seems too small.

Line 327: The OB group in current study is more like class 1 obese or just spread out in class 1-3 obese?
